# Immunogenicity and Antigenicity of Rabbit Hepatitis E Virus-Like Particles Produced by Recombinant Baculoviruses

**DOI:** 10.3390/v13081573

**Published:** 2021-08-09

**Authors:** Huimin Bai, Michiyo Kataoka, Yasushi Ami, Yuriko Suzaki, Naokazu Takeda, Masamichi Muramatsu, Tian-Cheng Li

**Affiliations:** 1Department of Basic Medicine and Forensic Medicine, Baotou Medical College, Baotou 014060, China; baihuimin16@163.com; 2Department of Pathology, National Institute of Infectious Diseases, Tokyo 208-0011, Japan; michiyo@nih.go.jp; 3Management Department of Biosafety, Laboratory Animal, and Pathogen Bank, National Institute of Infectious Diseases, Tokyo 208-0011, Japan; yami@nih.go.jp (Y.A.); ysuzaki@nih.go.jp (Y.S.); 4Research Institute for Microbial Diseases, Osaka University, Osaka 565-0781, Japan; seishunaotake@gmail.com; 5Department of Virology II, National Institute of Infectious Diseases, Tokyo 208-0011, Japan; muramatsu@nih.go.jp

**Keywords:** rabbit HEV, recombinant baculovirus, virus-like particles, VLPs, insect cells Tn5, vaccine

## Abstract

Rabbit hepatitis E virus (HEV) is a novel HEV belonging to genotype 3 (HEV-3) in the *Orthohepevirus A* species of the genus *Hepevirus*, family *Hepeviridae*. Rabbit HEV was originally isolated from rabbits and found to cause zoonotic infection. Although rabbit HEV can be successfully grown in culture with several cell lines, including the human carcinoma cell line PLC/PRF/5, it is difficult to obtain the large amounts of viral antigen required for diagnosis and vaccine development. In this study, we expressed N-terminal 13 and 111 aa-truncated rabbit HEV ORF2 proteins using recombinant baculoviruses and obtained two types of virus-like particles (VLPs), RnVLPs and RsVLPs with ~35 and 24 nm diameter, respectively. Anti-rabbit HEV IgG antibodies were induced in high titer by immunizing rabbits with RnVLPs or RsVLPs. The antibody secretion in the serum persisted more than three years. RsVLPs showed stronger antigenic cross-reactivity against HEV-1, HEV-3 and HEV-4 than rat HEV. Moreover, anti-RsVLPs antibodies neutralized not only the cognate virus but also HEV-1, HEV-3 and HEV-4 ex vivo, indicating that rabbit HEV had the same serotype as human HEVs. In contrast, the antibody did not block rat HEV infection, demonstrating that rat HEV belonged to a different serotype. Animal experiments indicated that immunization with either RnVLPs or RsVLPs completely protected the rabbits from challenge by rabbit HEV, suggesting that the VLPs are candidates for rabbit HEV vaccine development.

## 1. Introduction

Hepatitis E virus (HEV), the causative agent of hepatitis E, primarily transmits to humans via the fecal-oral route through contaminated drinking water and is a major health problem in many developing countries [1,2,3]. HEV has a large number of animal reservoirs, including monkeys, swine, wild boar, rabbits, camels, and rats, and is also recognized as an important emerging zoonotic virus causing acute and chronic liver disease in humans [4,5,6,7,8,9,10]. In fact, the cases of hepatitis E transmitted through contaminated food, such as raw or undercooked animal meat products, are increasing in both industrialized and developing countries [11].

HEV belongs to the family *Hepeviridae*, which includes two genera, *Orthohepevirus* and *Piscihepevirus* [12]. The genus *Orthohepevirus* includes four species, Orthohepevirus A–D. Orthohepevirus A consists of 8 genotypes (HEV-1 to -8), and HEV-1 to HEV-4 and HEV-7 are known to infect humans and cause acute or chronic hepatitis [12,13]. HEV-5 and HEV-8 have a potential risk of zoonotic infection because they are capable of infecting cynomolgus monkeys [14,15]. Although two strains of HEV-6 have been detected in wild boar, whether these viruses transmit to humans is unknown [16,17]. 

Following the first detection of a rabbit HEV in a farmed rabbit in China in 2009, many rabbit HEV strains have been detected in farmed rabbits, wild rabbits or specific pathogen-free rabbits worldwide, suggesting that rabbit HEV infection is common in rabbits and that rabbits are a natural host of rabbit HEV [18,19,20,21,22]. Because rabbit HEV is genetically closest to HEV-3, it was assigned to HEV-3ra, a subtype of HEV-3 [13]. The genome organization and characteristics of HEV-3ra are similar to those of other HEV genotypes of *Orthohepevirus A*. ORF1 encodes a non-structural polyprotein containing domains consistent with a methyltransferase, a peptide containing a Y-domain, a papain-like cysteine protease, a peptide with a hypervariable region (HVR), a helicase, and an RNA-dependent RNA polymerase (RdRp). ORF2 encodes a viral capsid protein. ORF3 encodes a phosphoprotein which interacts with the cellular cytoskeleton and is associated with virion release [7,23,24].

Rabbit HEV virus has been successfully transmitted to cynomolgus macaques by intravenous injection, demonstrating a potential cross-species transmission [25]. In fact, recent studies revealed that rabbit HEV can be transmitted from rabbits to humans and cause zoonotic infection, and that rabbit slaughterhouse workers are at high risk of HEV infection [26,27]. Therefore, the diagnosis and vaccine development for rabbit HEV stand in need of urgent attention. Although a cell culture system to grow the virus has been established, it is difficult to obtain the large amounts of antigen required for diagnosis and vaccine development using this system [28,29].

Recombinant baculovirus expression systems are a powerful tool for protein expression. We previously created an efficient system for the expression of N-terminal truncated capsid proteins of HEV-1, HEV-3 to HEV-7, ferret HEV and rat HEV that self-assembled into virus-like particles (VLPs) and obtained a large amount of purified VLPs [30,31,32,33,34]. To obtain a large amount of the rabbit HEV antigen, we expressed two types of the N-terminus-truncated ORF2s and obtained two different sizes of the VLPs. These VLPs exhibited antigenic cross-reactivity against HEV-1, HEV-3 and HEV-4, induced high titer antibody in rabbits, and were protected from a challenge by the cognate virus. 

## 2. Materials and Methods

### 2.1. Construction of Transfer Vectors

Two types of the truncated ORF2s of rabbit HEV were amplified by a polymerase chain reaction (PCR) using a plasmid, pUC57-T7RHEV-F, as a template containing the complete genome of a rabbit HEV strain (JQ013791) [29]. An N-terminal 13 aa-truncated ORF2 was amplified with a forward primer RFN13 (5′-AGGATCCATGCTGCCTATGCTGCCCGCGCCA-3′) and a reverse primer RCR (5′-CTCTAGATTAAGACTCCCGGGTTTTACCTA-3′), whereas an N-terminal 111-aa-truncated ORF2 was amplified with a forward primer RFN111 (5′-AGGATCCATGGCCGTTTCACCAGCCCCTGACA-3′) and the same reverse primer. The forward primers, RFN13 and RFN111, contained *Bam*HI sites (underlined) before the start codon and the reverse primer RCR contained an *Xba*I site (underlined) after the stop codon. The amplification was performed under the following conditions: 60 s incubation at 96 °C, followed by 35 cycles of 30 s at 95 °C, 30 s at 55 °C, and 90 s at 72 °C, and a final extension at 72 °C for 7 min. 

The PCR products were purified with a gel purification kit (Qiagen, Valencia, CA, USA) and cloned into TA 2.1 cloning vector (Invitrogen, San Diego, CA, USA), and HEV sequences were analyzed with M13 reverse and M13 (−20) forward primers. The plasmids were digested with *Bam*HI and *Xba*I, and the truncated ORF2s were ligated with a baculovirus transfer vector pVL1393 (Pharmingen, San Diego, CA, USA) to yield the plasmids pVL1393-Rn13ORF2F and pVL1393-Rn111ORF2F.

### 2.2. Construction of Recombinant Baculoviruses and Expression of Capsid Proteins 

An insect cell line, Sf9 (RIKEN Cell Bank, Tsukuba, Japan), was incubated at 26.5 °C in Grace’s insect medium (Gibco, Grand Island, NY, USA) supplemented with 10% FBS and 0.26% tryptose phosphate broth (BD, Sparks, MD, USA). The Sf9 cells were co-transfected with BaculoGold, a linearized wild-type Autographa californica nuclear polyhedrosis virus DNA (BD Biosciences, San Diego, CA, USA), and the transfer plasmid, pVL1393-Rn13ORF2F or pVL1393-Rn111ORF2F, by a Lipofectin-mediated method as specified by the manufacturer. The recombinant viruses were designated Ac[Rn13ORF2F] or Ac[Rn111ORF2F]. To achieve large-scale expression, an insect cell line from Trichoplusia ni, BTL-Tn 5B1-4 (Tn5), was infected with the recombinant baculoviruses at an m.o.i. of 10, and the cells were cultured in EX-CELL 405 medium (SAFC Biosciences, Lenexa, KS, USA) at 26.5 °C as described previously [30]. 

In addition, the recombinant baculoviruses containing the N-terminal 13 and 111 aa-truncated ORF2 of three other rabbit HEV strains, rbIM004 (AB740222), GDC9 (FJ906895) and GDC46 (FJ906896), were constructed in the same manner.

### 2.3. Purification of VLPs

The recombinant baculovirus-infected Tn5 cells were harvested on day 7 post-infection (p.i.). The cells and culture supernatants were separated by centrifugation at 10,000× *g* for 60 min. The supernatant was concentrated at 126,000× *g* for 3 h in a Beckman SW32Ti rotor (Beckman, Brea, CA, USA). The resulting pellet was resuspended in EX-CELL 405 medium at 4 °C overnight and then purified by CsCl gradient centrifugation. As a separate preparation, the cells were treated with a denaturation buffer containing 50 mM sodium borate, 150 mM NaCl, 1% Nonidet P-40, 0.5% sodium deoxycholate and 5% 2-mercaptoethanol, and gently rocked at room temperature for 2 h. The lysate was diluted with EX-CELL 405 medium, and then concentrated by centrifugation as described above. For CsCl gradient centrifugation, 4.5 mL of the samples was mixed with 2.1 g of CsCl and centrifuged at 100,000× *g* for 24 h at 10 °C in a Beckman SW55Ti rotor. The gradient was fractionated into 250-μL aliquots, and each fraction was weighed to estimate the buoyant density and isopycnic point. Each fraction was diluted with EX-CELL 405 medium and centrifuged for 2 h at 100,000× *g* in a Beckman TLA55 rotor (Beckman) to pelletize the VLPs. 

### 2.4. Transmission Electron Microscopy (TEM)

The purified VLPs were placed on a carbon-coated grid for 45 s, rinsed with distilled water, and stained with a 2% uranyl acetate solution. The grids were observed under a transmission electron microscope (HT7700; Hitachi High Technologies, Tokyo, Japan) at 80 kV.

### 2.5. Immunization of Rabbits with RsVLPs and RnVLPs

Two 15-week-old female Japanese white rabbits (SLC, Shizuoka, Japan), Rab-a and Rab-b, negative for anti-HEV antibodies and HEV RNA, were subcutaneously immunized with 100 µg of the RsVLPs and RnVLPs, respectively, and the booster injections were carried out on day 14 and 28 after the first injection with half-doses (50 µg) of the VLPs. All the injections, including the booster injections, were carried out without any adjuvant. The serum samples were collected on days 14, 28 (before the second and third immunization) and 42 p.i. and used for the detection of the anti-rabbit HEV IgG antibodies. 

### 2.6. Challenge of Rabbits and Sample Collection 

The challenge virus was derived from the fecal specimen of a rabbit, RY5, which was persistently infected with a rabbit HEV strain (LC484431) [29]. The fecal specimen collected from RY5 on day 168 p.i. was diluted with 10 mM PBS to prepare a 10% (*w*/*v*) stool suspension. The suspension was shaken at 4 °C for 1 h, then clarified by centrifugation at 10,000× *g* for 30 min. Finally, the suspension was passed through a 0.45-µm membrane filter (Millipore, Bedford, MA, USA). The RNA copy number of the challenge virus was 5 × 10^6^ copies/mL. 

The VLPs-immunized rabbits, Rab-a and Rab-b, were intravenously inoculated with 1 mL of the challenge virus through an ear vein. A cognate rabbit negative for the anti-rabbit HEV antibody, Rab-n, was inoculated with the same dose of the challenge virus to confirm the infectivity. 

The serum samples were collected weekly, and used for the determination of the viral RNA, anti-rabbit HEV IgG and IgM antibodies, and alanine aminotransferase (ALT) levels. The fecal specimens were collected two times per week and used for the detection of the viral RNA. The animal experiments were reviewed and approved by the institutional ethics committee and were performed according to the “Guides for Animal Experiments at the National Institute of Infectious Diseases, Tokyo” under codes 118044 (24 May 2018) and 120086 (8 October 2020). Rabbits were individually housed in Biosafety Level-2 facilities. 

### 2.7. Detection of Anti-Rabbit HEV IgG and IgM Antibodies 

An enzyme-linked immunosorbent assay (ELISA) for the detection of anti-rabbit HEV IgG or IgM antibodies using RsVLPs or RnVLPs as the antigen was performed as described previously [29,30]. Flat-bottomed 96-well polystyrene microplates were coated with 100 ng/well of the VLPs, and the duplicates of the 1:200 diluted serum samples were used. Horseradish peroxidase (HRP)-conjugated goat anti-rabbit IgG-heavy and light-chain antibody (1:5000) (Cappel, Westchester, PA, USA), and HRP-conjugated goat anti-rabbit IgM antibody (1:20,000) (abcam, Tokyo, Japan) were used to detect the anti-rabbit IgG and IgM antibodies, respectively. To determine the cut-off value of the IgG and IgM, 20 sera collected from SPF Japanese white rabbits were used. The optical density (OD) values of the IgG ranged from 0.006 to 0.123, and the cut-off value of 0.151 was obtained on the basis of the mean optical density plus 3 times the standard deviation (0.058 + 3 × 0.031). Similarly, the OD values of the IgM ranged from 0.045 to 0.154, and the cut-off value was calculated as 0.199 (0.097+ 3 × 0.034).

### 2.8. Titration of Rabbit IgG Antibodies

To examine the reactivity between various HEV VLPs and the corresponding monospecific anti-VLPs antibodies, an ELISA for the detection of the rabbit IgG was performed using the VLPs of HEV-1, HEV-3, HEV-4, rabbit HEV or rat HEV [30,31]. The duplicates of the rabbit sera were subjected to two-fold dilution starting from 1:200, and used to determine the antibody titers. The reciprocal of the highest dilution at the positive judgment was used for the antibody titer.

### 2.9. Detection of the Capsid Protein 

The HEV capsid protein was detected by an antigen-capture ELISA as described previously with slight modifications [30]. Briefly, a 96-well microplate was coated with 1:2000 diluted rabbit anti-HEV-1 VLPs serum to trap HEV-1, HEV-3, HEV-4 and rabbit HEV, or with rabbit anti-rat HEV VLPs serum to trap rat HEV [30,35,36]. The cell culture supernatant (100 µL) was used for the assay. An uninfected cell culture supernatant (three wells per plate) served as the negative control. The detection of the capsid protein was performed using guinea pig anti-HEV-1 VLPs or guinea pig anti-rat HEV VLPs hyperimmune serum (1:2000). HRP-conjugated goat anti-guinea pig IgG antibody (1:2000; Cappel, Durham, NC, USA) was used as the secondary antibody. When the ratio of the OD values between the sample and negative control was higher than 3.0, the sample was judged to be positive. 

### 2.10. Neutralizing Assay 

A cell culture-based neutralizing assay was used to examine the neutralizing activity of anti-rabbit HEV VLPs antibodies as described previously [35]. Five HEV strains were used for the neutralizing assay. The HEV-1 (subtype 1a) (LC061267) was derived from the stool specimen from a cynomolgus monkey, which had been experimentally infected with an Indian strain [37]. The HEV-3 (subtype 3k) (AB740232) was derived from a pig stool specimen in Japan [38]. HEV-4 (subtype 4i) (DQ079628) was derived from a wild boar caught in Japan [39]. Rat HEV (JX120573) was derived from a Vietnamese wild rat [40]. All the HEV strains were cultured in PLC/PRF/5 and the supernatants were used for the neutralizing assay. Briefly, pre- and RsVLPs-immunized rabbit sera were heated at 56 °C for 30 min and then diluted to 1:10 with Medium 199. One milliliter of the solution containing 2 × 10^6^ RNA copies of rabbit HEV, HEV-1, HEV-3, HEV-4 or rat HEV was mixed with 1 mL of the diluted antiserum and incubated at 37 °C for 1 h, and then at 4 °C for 3 h. The PLC/PRF/5 cells were cultured in 25 cm^2^ cell culture bottles (5 × 10^5^ cells/well) with 10 mL DMEM containing 10% (*v*/*v*) heat-inactivated FBS. After the cell culture medium was removed, 1 mL of the virus/serum mixture was added to each well. After adsorption at 37 °C for 1 h, the cells were washed three times with PBS, and 10 mL of maintenance medium consisting of medium 199 (Gibco, Grand Island, NY, USA), 2% (*v*/*v*) heat-inactivated FBS and 10 mM MgCl_2_ was added to each well. The culture medium was replaced with new medium every 4 days. The neutralizing activity was monitored by the detection of the HEV capsid protein in the cell culture supernatant at 4 weeks post-inoculation by ELISA. 

### 2.11. Quantitative Real-Time Reverse Transcription-Polymerase Chain Reaction (RT-qPCR) for Detection of Rabbit HEV RNA 

The viral RNA was extracted from 200 µL of the samples using a MagNA Pure LC system with a Total Nucleic Acid Isolation Kit (Roche Applied Science, Mannheim, Germany) according to the manufacturer’s recommendations. A one-step RT-qPCR was carried out with a 7500 FAST Real-Time PCR System (Applied Biosystems, Foster City, CA, USA) using TaqMan Fast Virus 1-step Master Mix (Applied Biosystems). The RT-qPCR was performed under a protocol of 5 min at 50 °C, 20 s at 95 °C, followed by 40 cycles of 3 s at 95 °C and 30 s at 60 °C using the primer pair of forward primer 5′-GGTGGTTTCTGGGGTGAC-3′ (nt 5346–5363) and reverse primer 5′-AGGGGTTGGTTGGATGAA-3′ (nt 5393–5415), and the probe 5′-FAM- TGATTCTCAGCCCTTCGC-TAMRA-3′ (nt 5369–5386) [41]. A 10-fold serial dilution of the capped rabbit RHEV RNA (10^7^ to 10^1^ copies) was used as the standard for quantitation of the viral genome copy numbers. Amplification data were collected and analyzed with Sequence Detector software ver. 1.3 (Applied Biosystems).

### 2.12. Examination of the Alanine Aminotransferase (ALT) Levels 

ALT values in the monkey sera were monitored weekly using a Fuji Dri-Chem Slide GPT/ALT-PIII kit (Fujifilm, Saitama, Japan). The geometric mean titers of ALT over the preinoculation period were defined as normal ALT, and a 2-fold or greater increase at the peak was considered a sign of hepatitis [29].

## 3. Results 

### 3.1. The N-terminal 111 aa-Truncated ORF2 Protein Formed Small VLPs 

A recombinant baculovirus comprising the N-terminal 111 aa-truncated ORF2 of rabbit HEV, Ac[Rn111ORF2F], was inoculated on Tn5 cells and incubated for 7 days. The supernatants were collected, concentrated, and subjected to a CsCl gradient centrifugation. A 53 kDa protein (p53) band that processed from N-terminal 111 aa-truncated ORF2 of rabbit HEV was primarily distributed in fractions 12 to 16 with an average density of 1.285g/cm^3^ (range: 1.284 g/cm^3^ to 1.289g/cm^3^) (Figure 1a). Electron micrographs revealed many spherical particles with a diameter of approx. 24 nm throughout the fractions; Figure 1b shows the particles in fraction 14. The size of the VLPs was similar to the sizes of HEV-1 and HEV-3 to HEV-7 produced with the recombinant baculoviruses harboring the same deletion in ORF2. The yield of the purified VLPs reached 0.78 mg per 10^7^ cells. These results indicated that p53 produced by expression of the N-terminal 111 aa-truncated rabbit HEV ORF2 self-assembled into VLPs. The VLPs produced by Ac[Rn111ORF2F] were designated RsVLPs. 

### 3.2. The N-terminal 13 aa-Truncated ORF2 Protein Formed Native Virion-Sized VLPs

We previously demonstrated that the expression of the N-terminal 13 aa-truncated ORF2 of HEV-3, HEV-5, HEV-6 and HEV-7 with the corresponding recombinant baculovirus resulted in the formation of self-assembled VLPs that retained a similar diameter to the native virus particles, although the VLPs were not released into the supernatants [32,34,42]. The Ac[Rn13ORF2F]-infected Tn5 cells were incubated for 7 days, and then the supernatant was removed, and the cells were treated with the denaturing buffer as described in the Materials and Methods. After CsCl gradient centrifugation, a 64 kDa protein (p64) mainly appeared in fractions 9 to 13 with an average density of 1.300 g/cm^3^ (Figure 2a). The observation of fractions 9 to 13 by electron micrograph revealed spherical particles with a diameter of approx. 35 nm; Figure 2b shows the particles in fraction 11. The size of the particles produced by Ac[Rn13ORF2F] was larger than that of the particles generated by Ac[Rn111ORF2F] and similar to that of the native HEV particles. We designated these particles RnVLPs. The yield of the purified RnVLPs was 0.23 mg per 10^7^ cells. In addition to p64, p53 was detected in fractions 14 to 18 with an average density of 1.285 g/cm^3^. The electron micrograph of fraction 16 showed small VLPs with a diameter of approx. 24 nm (Figure 2c), indicating that the Ac[Rn13ORF2F]-infected cells produced both virion-size and small VLPs. In addition to p64 and p53, a 40 kDa protein (p40) was observed in fractions 4 and 5 with an average density of 1.350 g/cm^3^. This protein seemed to be the coat protein of the nodavirus, which was spontaneously induced by infection with baculoviruses as described in our previous study; the p40 protein was not released into the supernatant [43].

We employed the same method to express the capsid protein using three other rabbit HEV strains, i.e., rbIM004, GDC9 and GDC46, but the VLPs were exclusively produced by the N-terminal 13 and 111 aa-truncated ORF2s derived from one of these strains: rbIM004. 

### 3.3. Immunogenicity of Rabbit HEV VLPs 

To examine the immunogenicity of the VLPs, we immunized Rab-a with RsVLPs and Rab-b with RnVLPs as described in the Materials and Methods. The serum IgG antibody titers were 1:6400 in Rab-a and 1:51,200 in Rab-b on day 14 p.i. (immediately before the second immunization), and they increased to 1:102,400 on day 28 p.i. (immediately before the third immunization) in both rabbits and reached 1:204,800 on day 42 p.i. (two weeks after the third immunization) in both rabbits (Figure 3). The serum IgG antibody titers of Rab-a determined with RnVLPs as the antigen and those of Rab-b determined with RsVLPs as the antigen showed the same reaction patterns, suggesting that the antigenicity of RsVLPs and that of RnVLPs were similar. These results indicated that both the small VLPs and the native virion size VLPs induced strong immune responses in rabbits.

### 3.4. RsVLPs-Immunized Rabbit Serum Has Neutralizing Activity against HEV Infection Ex Vivo

To examine the neutralizing activity of the VLPs-immunized serum, we mixed the RsVLPs-immunized rabbit serum with HEV-1, HEV-3, HEV-4, rabbit HEV or rat HEV, and then incubated the resulting mixture and used it to inoculate PLC/PRF/5 cells. Preimmune rabbit serum was used as the negative control. The cell culture supernatants were collected on day 28 p.i. and the corresponding HEV capsid protein was detected by the antigen-capture ELISA. As shown in Figure 4, the HEV-1, HEV-3, HEV-4 and rabbit HEV capsid proteins were detected in the samples inoculated with preimmune rabbit serum with OD values of 0.848, 0.998, 0.894 and 0.751, respectively, but the capsid protein was not detected when these samples were incubated with anti-RsVLPs serum, indicating that the anti-RsVLPs antibody was capable of neutralizing not only homologous rabbit HEV but also heterologous HEV-1, HEV-3 and HEV-4. Therefore, the rabbit HEV belongs to the same serotype as HEV-1, HEV-3 and HEV-4.

In contrast, the rat HEV capsid proteins were detected in both the preimmune and anti-RsVLPs serum with OD values of 0.664 and 0.652, respectively, suggesting that the serotype of rat HEV was different from that of rabbit HEV. 

### 3.5. Anti-RsVLPs Sera Cross-Reacted with Other Genotypes of HEVs

To further explore the antigenic cross-reactivity among rabbit HEV, HEV-1, HEV-3, HEV-4 and rat HEV, we performed antibody ELISAs to compare the reactivity between homologous and heterologous immune sera (Figure 5). The anti-HEV-1 IgG antibody titer against the homologous HEV-1 antigen was 1:3,276,800, and that against the heterologous rabbit HEV antigen was identical at 1:3,276,800 (Figure 5a). Similar reaction patterns were observed when the anti-HEV-3 IgG antibody (Figure 5b) and the anti-HEV-4 IgG antibody (Figure 5c) titers were determined. In contrast, the reactivity of the anti-rat HEV antibody against the rabbit HEV was 1:51,200, although that against the cognate rat HEV antigen was as high as 1:204,800 (Figure 5d). When the reactivity of the anti-rabbit HEV IgG was examined, the titer against the cognate rabbit HEV (1:204,800) was similar to those against HEV-1 (1:102,400), HEV-3 (1:204,800) and HEV-4 (1:102,400) (Figure 5e). However, the titer against the rat HEV was 1:12,800, significantly lower than 1:204,000, the titer against the cognate rabbit HEV (Figure 5e). These results suggested that the antigenicity of rabbit HEV was more similar to those of HEV-1, HEV-3 and HEV-4 HEV than that of rat HEV. 

### 3.6. RsVLPs- and RnVLPs-Immunized Rabbits Were Protected from Rabbit HEV Infection

The RsVLPs-immunized Rab-a and RnVLPs-immunized Rab-b were further housed separately, and the titers of the anti-rabbit HEV serum IgG at month 39 p.i. were monitored. The IgG titers were maintained at 1:3200 in Rab-a and 1:6400 in Rab-b. When these rabbits were boosted with 25 µg of RsVLPs (Rab-a) or RnVLPs (Rab-b), the IgG antibody titers increased to 1:204,800 in two weeks. 

Next, to explore whether these rabbits were protected from a challenge with the cognate rabbit HEV, Rab-a and Rab-b were intravenously inoculated with 1 mL of the rabbit HEV (2 × 10^7^ copies/mL) through an ear vein at 3 weeks after the boost immunization. A rabbit, Rab-n, negative for the rabbit HEV RNA and anti-rabbit HEV antibodies, was inoculated with the same amount of the rabbit HEV and used as a control to evaluate the infectivity of the challenge virus. The viral RNA was detected in fecal specimens from Rab-n at day 7 p.i. with 1.96 × 10^4^ copies/g, and the viral RNA reached a peak at day 14 p.i. with a copy number of 9.85 × 10^6^ copies/g, and then decreased gradually and became undetectable from day 52 p.i., whereas it was detected in the serum samples at days 14 and 21 p.i., at 3.29 × 10^3^ and 2.80 × 10^3^ copies/mL (Figure 6a). The anti-rabbit HEV IgG and IgM antibodies were detected in the sera on day 21 p.i., with OD values of 0.562 and 0.287, respectively. The IgG antibody increased to above 3.0 and remained at the same level until day 77 p.i., and the IgM antibody increased to a peak at day 28 p.i. with an OD value of 0.852, and then decreased gradually (Figure 6b). Although no significant ALT elevations were observed in Rab-n, the results clearly confirmed that the infection occurred, and the virus used for the inoculation was infectious (Figure 6c). 

In contrast, no viral RNAs were detected in the fecal or serum specimens of Rab-a or Rab-b (Figure 6d,g). The anti-HEV IgG antibodies were maintained at high levels before and after the challenge, but no anti-IgM antibodies were detected (Figure 6e,h). No ALT elevation was observed in either of these rabbits until day 77 p.i. (Figure 6f,i). These results revealed that the rabbits immunized with the RsVLPs and RnVLPs were completely protected from rabbit HEV infection.

## 4. Discussion

Rabbit HEV is a novel agent of zoonotic infection and has become distributed worldwide, making diagnosis and vaccine development for rabbit HEV an urgent matter. Since the growth ability of rabbit HEV in cell culture is extremely low, an expression system capable of producing a large amount of the viral antigen is necessary. Expression of the structural proteins by recombinant baculoviruses has long been used to generate VLPs of various viruses, and VLPs produced by this system usually retain the immunogenic and physicochemical properties of the native virions. In addition, a large amount of purified VLPs of HEV-1, and HEV-3 to -7 have been obtained [30,32,34,42]. We employed a recombinant baculovirus system to express the N-terminal 13 and 111 aa-truncated rabbit HEV ORF2 and obtained virion-size 35 nm VLPs (RnVLPs) and small 24 nm VLPs (RsVLPs), respectively. The RsVLPs were released into the culture supernatants, making their purification easier. In contrast, the RnVLPs were retained in the infected cells, so that a denaturation buffer was needed during the purification. The yield of the purified RsVLPs reached 0.78 mg/10^7^ cells and was higher than that of RnVLPs (0.23 mg/10^7^ cells). The size of the RnVLPs was similar to that of the native virion, and the structure analyses might be useful to understand the mechanism of the virion assembly. 

We expressed the capsid protein of two other rabbit HEV strains, GDC9 (FJ906895) and GDC46 (FJ906896), the first identified rabbit HEVs [7]. However, neither the N-terminal 111 nor the 13 aa-truncated ORF2 of GDC9 and GDC6 generated VLPs using the same protocol. We previously observed that one amino acid mutation (M358T) significantly affected the assembly of the N-terminal 111 aa-truncated HEV-7 ORF2 into the small VLPs [34], which suggests that some key mutation(s) might be present in these two strains. 

HEVs are generally thought to represent a single serotype [44,45]. However, over the last decade many novel strains of HEV have been identified in several animal species, and the genus *Orthohepevirus* was shown to include at least four species. It is important to clarify the serotype(s) among HEV species in the genus *Orthohepevirus* of the family *Hepeviridae*. Our previous studies confirmed that HEV-3, HEV-5 and HEV-7 were neutralized by the anti-HEV-1, -HEV-3 and -HEV-4 antibodies, but not by anti-rat HEV antibody, suggesting that the serotype of rat HEV is different from that of HEV-1, HEV-3, HEV-4, HEV-5 and HEV-7 [14,32,35]. In the present study, we further confirmed that the anti-rabbit HEV antibodies neutralized HEV-1, HEV-3 and HEV-4 HEV ex vivo, and blocked infection in vivo, whereas anti-rabbit HEV antibodies did not neutralize rat HEV ex vivo, again underscoring the distinctness of the serotype of rat HEV. In fact, rat HEV ORF2 shared less than 60% aa sequence identities with HEV-1 to HEV-7.

When rabbits were immunized with RsVLPs or RnVLPs, a strong immune response was induced in the absence of any adjuvant, suggesting that both VLPs were highly immunogenic. Moreover, the RsVLPs- and RnVLPs-immunized rabbits were protected from the rabbit HEV infection, strongly suggesting that the VLPs are candidates for HEV vaccine development.

## Figures and Tables

**Figure 1 viruses-13-01573-f001:**
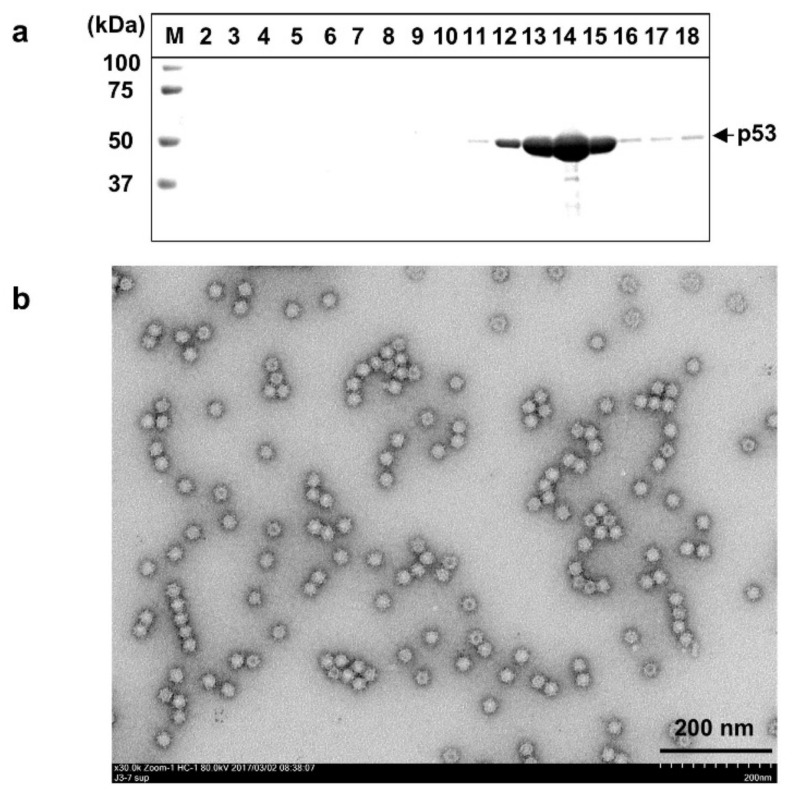
Purification of RsVLPs. The supernatant of Ac[Rn111ORF2]-infected Tn5 cells was collected on day 7 p.i. and concentrated, and then purified by CsCl equilibrium density gradient centrifugation. Aliquots from each fraction were analyzed by electrophoresis on 5–20% polyacrylamide gel and stained with Coomassie Brilliant Blue (CBB) (**a**). RsVLPs in fraction 14 were observed by electron microscopy (EM) (**b**). Bar: 200 nm. M: molecular weight maker.

**Figure 2 viruses-13-01573-f002:**
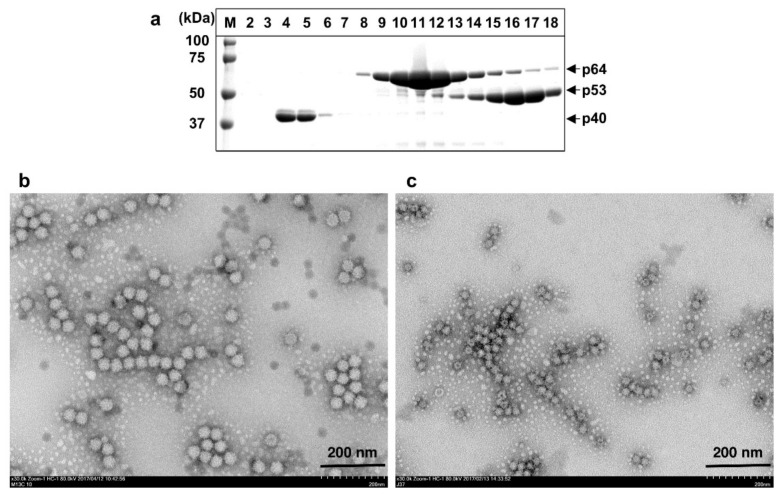
Purification of RnVLPs. Ac[Rn13ORF2]-infected Tn5 cells were harvested on day 7 p.i. and treated with denaturation buffer, and then purified by CsCl equilibrium density gradient centrifugation. Aliquots from each fraction were analyzed by electrophoresis on 5–20% polyacrylamide gel and stained with CBB (**a**). RnVLPs (**b**) in fraction 11 and RsVLPs (**c**) in fraction 16 were observed by EM. Bar: 200 nm. M: molecular weight maker.

**Figure 3 viruses-13-01573-f003:**
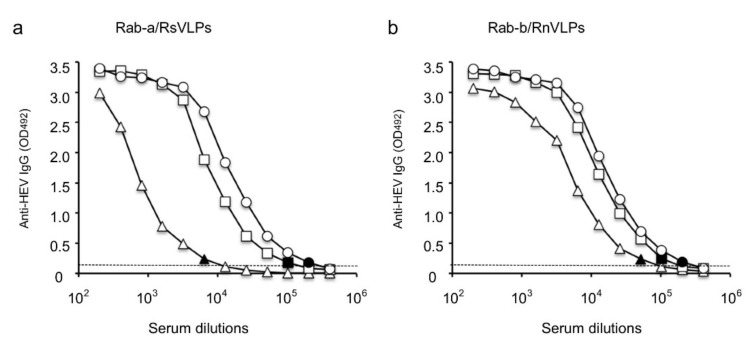
Immunogenicity of RsVLPs and RnVLPs. Two rabbits, Rab-a (**a**) and Rab-b (**b**), were immunized with RsVLPs and RnVLPs, respectively, and the sera were collected on day 14 (△), 28 (☐) and 42 (◯) p.i. Flat-bottomed 96-well polystyrene microplates were coated with 100 ng/well of the RsVLPs or RnVLPs and incubated with the corresponding, serially diluted rabbit sera. The IgG antibodies elicited on each day were measured by ELISA. Dotted lines indicate the cut-off value. The black shapes indicate the endpoints of the IgG antibody titers.

**Figure 4 viruses-13-01573-f004:**
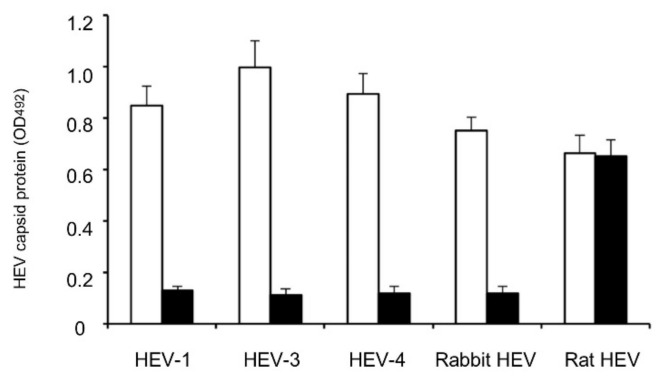
Neutralizing activity of anti-RsVLPs antibody. The neutralizing activity of anti-RsVLPs antibody was examined using a cell culture-based neutralization test. Different genotypes of the HEV, i.e., HEV-1, HEV-3, HEV-4 and rat HEV, and cognate rabbit HEV were used. A mixture of 2 × 10^6^ RNA copies of the HEV and 1:10 diluted anti-RsVLPs serum was incubated and used to inoculate PLC/PRF/5 cells. After incubation for 4 weeks, the capsid protein produced in the supernatant was measured by an antigen-capture ELISA. Triplicate samples were used for each combination. Bars indicate the OD values of HEV capsid protein. White bars: preimmune rabbit serum; Black bars: rabbit anti-RsVLPs serum.

**Figure 5 viruses-13-01573-f005:**
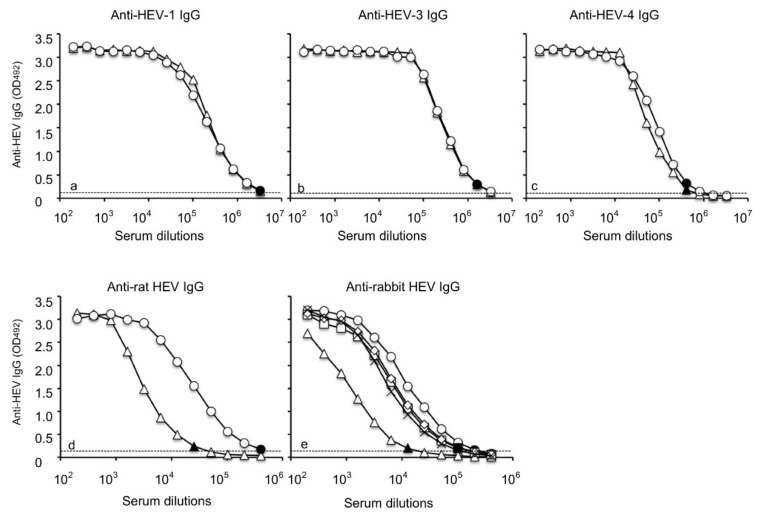
Antigenic cross-reactivity among HEV-1, HEV-3, HEV-4, rabbit HEV and rat HEV. The reactivity of anti-HEV-1 IgG to the homologous antigen HEV-1 VLPs (◯) and heterologous antigen rabbit HEV (△) was examined by antibody ELISA (**a**). Similarly, the reactivity of anti-HEV-3 IgG (**b**), that of anti-HEV-4 IgG (**c**) and that of anti-rat HEV IgG (**d**) was examined. The reactivity of anti-rabbit HEV IgG was determined by an antibody ELISA using the heterologous VLPs of HEV-1 (☐), HEV-3 (◇), HEV-4 (✕) and rat HEV (△) and was compared with that of homologous rabbit HEV VLPs (RsVLPs) (◯) (**e**). Dotted lines indicate the cut-off values. The black shapes indicate the endpoints of the IgG antibody titers.

**Figure 6 viruses-13-01573-f006:**
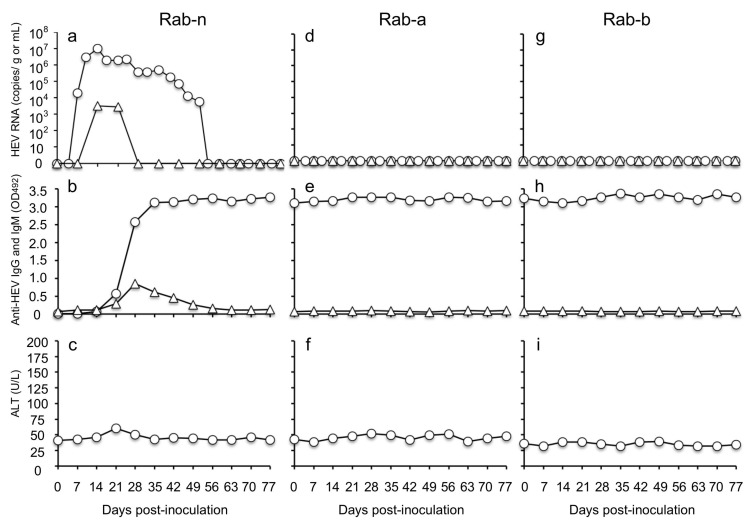
VLPs-immunized rabbits were protected from rabbit HEV infection. The RsVLPs-immunized rabbit, Rab-a, RnVLPs-immunized rabbit, Rab-b, and a naive rabbit, Rab-n, were intravenously inoculated with rabbit HEV. The kinetics of the RNA (**a**,**d**,**g**), anti-rabbit HEV IgG and IgM antibodies (**b**,**e**,**h**), and ALT (**c**,**f**,**i**) were measured. The rabbit HEV RNA copy numbers in the feces (◯) and sera (△) were determined by RT-qPCR. Anti-rabbit HEV IgG (◯) and IgM (△) in sera were determined by ELISA using RsVLPs as the antigen. ALT (◯) in the rabbit sera was determined by a commercial kit.

## Data Availability

The data that support the findings of this study are available from the manuscript.

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
