# Peer review of "Immunogenicity and Antigenicity of Rabbit Hepatitis E Virus-Like Particles Produced by Recombinant Baculoviruses"

_viruses, 2021, doi:10.3390/v13081573_

Round 1
Reviewer 1 Report
Bai et al. presented an interesting topic entitled"Immunogenicity and Antigenicity of Rabbit Hepatitis E Virus-Like Particles Produced by Recombinant Baculoviruses". The authors produced two types of the truncated ORF2s of rabbit HEV, designated RsVLPs and RnVLPs. The authors then tested the immunogenicity of these VLPs in rabbits and showed that they are immunogenic. Then the authors tested the neutralizing activity of the antibodies produced against VLPs both in vitro (using PLC/PRF/5 cells) and in vivo.
Overall the group of Prof Dr. Tian-Cheng Li is well known, and the study has an acceptable flow. I have some questions
1-Figure 4: Neutralizing activity of the anti-RsVLPs antibody. The authors tested RsVLPs-immunized rabbit serum against HEV-1, HEV-3, HEV-4, rabbit HEV, or rat HEV by measuring HEV capsid protein produced in the supernatant.
a) What is the status of HEV RNA in these experiments
b)What is the status of intracellular HEV ORF2 Ag
c) Please add the statistics
2- Figure 5: Antigenic cross-reactivity among HEV-1, HEV-3, HEV-4, rabbit HEV, and rat HEV. The authors want to verify that the antigenicity of rabbit HEV was more similar to those of HEV-1, HEV-3, and HEV-4 HEV than that of rat HEV. I guess this figure needs more clarification
a) What are the sources, subtypes of HEV-1, HEV-3, and HEV-4 isolates used in the comparison?
b) What is the source of Rat HEV used? Recent studies showed that rat HEV is infectious to humans.
c) The methodology and/references for VLPs of HEV-1, HEV-3, and HEV-4 are either missing or incomplete.
3) Figure 6: VLPs-immunized rabbits were protected from rabbit HEV infection. There are some missing information
a) how many rabbits used/ group?
b) Did the author try to prove the neutralization activity of anti-RSVLP sera after animal challenge with heterologous virus
Minor changes
1- Introduction: Page 1 line39-40: HEV has a large number of animal reservoirs, including monkeys, swine, wild boar, rabbits, camels, and rats, and is also recognized as an important emerging zoonotic virus causing acute and chronic liver disease in humans
Please include that cows and goats (ruminants) are potential sources of HEV infection (PMID: 31874303, PMID: 32659521)
2- Methodology" page 4 lone 181-182: The HEV capsid protein was detected by an antigen-capture ELISA as described previously with slight modifications. Please add the reference
3- Discussion: It is slightly short, but it is ok. Can the authors discuss that this potential neutralizing antibody can decrease the development of extrahepatic disorders?
Author Response
Responses to Reviewer 1:
Reviewers' comments:
Bai et al. presented an interesting topic entitled "Immunogenicity and Antigenicity of Rabbit Hepatitis E Virus-Like Particles Produced by Recombinant Baculoviruses". The authors produced two types of the truncated ORF2s of rabbit HEV, designated RsVLPs and RnVLPs. The authors then tested the immunogenicity of these VLPs in rabbits and showed that they are immunogenic. Then the authors tested the neutralizing activity of the antibodies produced against VLPs both in vitro (using PLC/PRF/5 cells) and in vivo.
Overall the group of Prof Dr. Tian-Cheng Li is well known, and the study has an acceptable flow. I have some questions. 1-Figure 4: Neutralizing activity of the anti-RsVLPs antibody. The authors tested RsVLPs-immunized rabbit serum against HEV-1, HEV-3, HEV-4, rabbit HEV, or rat HEV by measuring HEV capsid protein produced in the supernatant.
- What is the status of HEV RNA in these experiments,
Response:
Although the neutralizing activity was based on the detection of the HEV capsid protein in the cell culture supernatants, the HEV RNA and capsid protein showed similar pattern in the culture supernatants.
- What is the status of intracellular HEV ORF2 Ag
Response:
When HEV grows in infected cells, the capsid protein appears in the cell culture supernatants; therefore, we did not examine the level of the intracellular capsid protein.
- Please add the statistics
Response:
Because the aim of this experiment was to examine the neutralizing activity, we think it is appropriate to present the results without statistics.
2- Figure 5: Antigenic cross-reactivity among HEV-1, HEV-3, HEV-4, rabbit HEV, and rat HEV. The authors want to verify that the antigenicity of rabbit HEV was more similar to those of HEV-1, HEV-3, and HEV-4 HEV than that of rat HEV. I guess this figure needs more clarification
- a) What are the sources, subtypes of HEV-1, HEV-3, and HEV-4 isolates used in the comparison?
Response:
The HEV-1 (1a) was derived from the stool specimen of a cynomolgus monkey which had been experimentally infected with an Indian strain. The HEV-3 (3k) was derived from a pig stool specimen in Japan. HEV-4 (4i) was collected from a wild boar caught in Japan. Rat HEV (JX120573) was derived from a Vietnamese wild rat. All the HEV strains were cultured in PLC/PRF/5 and the supernatants were used for the neutralizing assay. This information was added to the revised manuscript on pages 9 to10, lines 219 to 226.
- b) What is the source of Rat HEV used? Recent studies showed that rat HEV is infectious to humans.
Response:
Rat HEV (JX120573) was derived from a Vietnamese wild rat. Whether this strain transmits to humans is unknown.
c) The methodology and/references for VLPs of HEV-1, HEV-3, and HEV-4 are either missing or incomplete.
Response:
The references for VLPs of HEV-1, HEV-3, and HEV-4 were cited in the Introduction (page 4, line 90).
3) Figure 6: VLPs-immunized rabbits were protected from rabbit HEV infection. There are some missing information
- how many rabbits used/ group?
Response:
The information on the rabbits used in this experiment was described in the section “2.6. Challenge of rabbit and sample collection” in the Materials and Methods.
- b) Did the author try to prove the neutralization activity of anti-RsVLP sera after animal challenge with heterologous virus
Response:
We would like to do this experiment in the future.
Minor changes
1- Introduction: Page 1 line39-40: HEV has a large number of animal reservoirs, including monkeys, swine, wild boar, rabbits, camels, and rats, and is also recognized as an important emerging zoonotic virus causing acute and chronic liver disease in humans
Please include that cows and goats (ruminants) are potential sources of HEV infection (PMID: 31874303, PMID: 32659521)
Response:
Since there is no direct evidence of the transmission from cows and goats to humans, and whether the cows were susceptible to HEV infection is unclear, we did not cited these references.
2- Methodology" page 4 lone 181-182: The HEV capsid protein was detected by an antigen-capture ELISA as described previously with slight modifications. Please add the reference
Response:
The reference was added.
3- Discussion: It is slightly short, but it is ok. Can the authors discuss that this potential neutralizing antibody can decrease the development of extrahepatic disorders?
Response:
While this question is certainly of interest, at present we do not have any information on this topic.

Reviewer 2 Report
The manuscript entitled “Immunogenicity and Antigenicity of Rabbit Hepatitis E Virus-Like Particles Produced by Recombinant Baculoviruses” by Bai et al sent for publication to Viruses, is focusing on very important nowadays topics like vaccine production in alternative expression systems. The manuscript will be of interest for the scientific community working in that area and would help them to update knowledge of HEV infection in animals. However, the authors should address to minor revise the manuscript in order to make it easy to understand. I left my comments and suggestions as follows.
Introduction:
The introduction is well written, the references used are up to date, and introduce the main objective.
-P70 It would be good to compare the expression systems used for the production of ORF 2 capsid protein and to highlight the advantages of a recombinant baculovirus system.
Materials and methods:
Materials and methods described well all protocols and allowed reproduction of the experiments.
-P97 Have you sequenced the obtained PCR products?
Also, when quoting companies it is necessary to add the full address of the company, including the country, p93, p94, p96, p100….
P124 should be 250 µL
P140 Challenge of rabbits - It would be good to explain here the method for the detection of viral RNA in the sample with which the experimental animals are infected
P173 2.8. Titration of rabbit IgG antibodies - more detailed explanations are needed
I don’t see the protocol for the alanine aminotransferase (ALT) levels checking!
Results:
Results are well presented, but I have some suggestions.
I suggest changing 3.1. with 3.2
3.1. The N-terminal 13 aa-truncated ORF2 protein formed native virion-sized VLPs
3.2. The N-terminal 111 aa-truncated ORF2 protein formed small VLPs
P224-225 these details are more about Materials and Мethods. Please, rewrite it.
P226 I would like more dilates for a 53 kDa protein (p53) band. If you are not familiar with the expression of ORF2, it is difficult to perceive the direct appearance of p53
P239 Aliquots from each fraction were analyzed by electrophoresis on 5%–20% polyacrylamide gel, and stained with Coomassie Brilliant Blue (CBB) (a).
It is better to be - (a) Aliquots from each fraction were analyzed by electrophoresis on 5%–20% polyacryla mide gel, and stained with Coomassie Brilliant Blue (CBB).
P240 (b) RsVLPs in fraction 14 were observed 240 by electron microscopy (EM).
P249 It would be good to explain the expression of ORP2 and the proteins that are synthesized from the recombinant expression system.
P260. Do you try to make a EM of fraction 4 (p40)?
P267 Figure 2. The explanation of (c ) is missing. The same suggestion as Figure 1.
P363 How would you comment on the lack of change in ALT in the presence of viral infection in control animals?
Discussion:
The discussion can be improved.
P385 It is not correct to comment on the morphology of particles only by their size. I suggest you paraphrase the expression.
P388-394 Please revise it. It is difficult to be understood.
My recommendation is to slightly expand the discussion and comment on the expression of ORF2 in different expression systems.
Overall, the manuscript is important for vaccine production progress and I would like to congratulate the authors for the good job.

Author Response
Responses to Reviewer 2:
Reviewers' comments:
The manuscript entitled “Immunogenicity and Antigenicity of Rabbit Hepatitis E Virus-Like Particles Produced by Recombinant Baculoviruses” by Bai et al sent for publication to Viruses, is focusing on very important nowadays topics like vaccine production in alternative expression systems. The manuscript will be of interest for the scientific community working in that area and would help them to update knowledge of HEV infection in animals. However, the authors should address to minor revise the manuscript in order to make it easy to understand. I left my comments and suggestions as follows.
Introduction: The introduction is well written, the references used are up to date, and introduce the main objective.
-P70 It would be good to compare the expression systems used for the production of ORF 2 capsid protein and to highlight the advantages of a recombinant baculovirus system.
Response:
We added the following sentence to describe the advantages of using a recombinant baculovirus system for expression of the HEV capsid proteins: “We previously created an efficient system for the expression of N-terminal truncated capsid proteins of HEV-1, HEV-3 to HEV-7, ferret HEV and rat HEV that self-assembled into virus-like particles (VLPs) and obtained a large amount of purified VLPs.” (page 4, lines 87 to 90)
Materials and methods:
Materials and methods described well all protocols and allowed reproduction of the experiments.
-P97 Have you sequenced the obtained PCR products? Also, when quoting companies it is necessary to add the full address of the company, including the country, p93, p94, p96, p100….
Response:
Yes, all the PCR products were confirmed by sequence analyses. This description was added on page 5, line 113, in the revised manuscript. We also added “USA” to the suppliers where appropriate.
P124 should be 250 µL
Response:
Thank you for pointing this out. We corrected it.
P140 Challenge of rabbits - It would be good to explain here the method for the detection of viral RNA in the sample with which the experimental animals are infected
Response:
The method for detection of HEV RNA was described in section 2.11, and the challenge virus was described in the same section.
P173 2.8. Titration of rabbit IgG antibodies - more detailed explanations are needed
Response:
The method and antigen used for the ELISA, along with the dilution and titer judgment were described in this section.
I don’t see the protocol for the alanine aminotransferase (ALT) levels checking!
Response:
We added this information as a new section, “2.12. Examination of the alanine aminotransferase (ALT) levels”, on page 11, lines 254 to 258 in the revised manuscript.
Results:
Results are well presented, but I have some suggestions.
I suggest changing 3.1. with 3.2
3.1. The N-terminal 13 aa-truncated ORF2 protein formed native virion-sized VLPs
3.2. The N-terminal 111 aa-truncated ORF2 protein formed small VLPs
Response:
Since we first obtained the small VLPs from the cell culture supernatants by infecting with recombinant baculovirus, Ac[Rn111ORF2F], we would like this sequence of events to be reflected in the presentation of subsections. That is, we would like present the section on “N-terminal 111 aa-truncated ORF2 protein formed small VLPs” first.
P224-225 these details are more about Materials and Мethods. Please, rewrite it.
Response:
The sentence was revised as follow: “A recombinant baculovirus comprising the N-terminal 111 aa-truncated ORF2 of rabbit HEV, Ac[Rn111ORF2F], was inoculated on Tn5 cells and incubated for 7 days.” (page 11, lines 262 to 263)
P226 I would like more dilates for a 53 kDa protein (p53) band. If you are not familiar with the expression of ORF2, it is difficult to perceive the direct appearance of p53.
Response:
We changed the phrase “53 kDa protein (p53) band” to “A 53 kDa protein (p53) band that processed from N-terminal 111 aa-truncated ORF2 of rabbit HEV . . . ” (page 11, lines 265 to 266)
P239 Aliquots from each fraction were analyzed by electrophoresis on 5%–20% polyacrylamide gel, and stained with Coomassie Brilliant Blue (CBB) (a).
It is better to be - (a) Aliquots from each fraction were analyzed by electrophoresis on 5%–20% polyacryla mide gel, and stained with Coomassie Brilliant Blue (CBB).
Response:
Thanks for the suggestion, but we would like to panel letters at the end of the sentences.
P240 (b) RsVLPs in fraction 14 were observed by electron microscopy (EM).
Response:
We would like to keep the panel letters at the end of the sentences.
P249 It would be good to explain the expression of ORF2 and the proteins that are synthesized from the recombinant expression system.
Response:
Since Ac[Rn13ORF2F] was defined as a recombinant baculovirus, we do not think that it is necessary to explain the expression of ORF2 and the proteins that are synthesized from the recombinant expression system.
P260. Do you try to make a EM of fraction 4 (p40)?
Response:
Yes, we observed the virus particles in fraction 4 by EM. The virus particles were similar to those of nodavirus, as was observed in our previous study (Li et al. 2007.JVI).
P267 Figure 2. The explanation of (c) is missing.
Response:
We added an explanation for panel (c) on page 14, line 315 in the revised manuscript.
The same suggestion as Figure 1.
Response:
We would like to keep the panel letters at the end of the sentences.
P363 How would you comment on the lack of change in ALT in the presence of viral infection in control animals?
Response:
Rabbit HEV infection induced not only acute hepatitis but also sub-clinical and persistent infection in rabbits. In the present study, rabbit HEV caused sub-clinical infection in the control animal without ALT elevation.
Discussion:
The discussion can be improved.
P385 It is not correct to comment on the morphology of particles only by their size. I suggest you paraphrase the expression.
Response:
We changed “morphology” to “size”.
P388-394 Please revise it. It is difficult to be understood.
Response:
The sentence was revised as follows: “We expressed the capsid protein of two other rabbit HEV strains, GDC9 (FJ906895) and GDC46 (FJ906896), the first identified rabbit HEVs [7]. However, neither the N-terminal 111 nor the 13 aa-truncated ORF2 of GDC9 and GDC6 generated VLPs using the same protocol,” (page 22, lines 452 to 455)
My recommendation is to slightly expand the discussion and comment on the expression of ORF2 in different expression systems.
Response:
According to the reviewer’s suggestion, we added the following passage to expand the discussion: “Expression of the structural proteins by recombinant baculoviruses has long been used to generate VLPs of various viruses, and VLPs produced by this system usually retain the immunogenic and physicochemical properties of the native virions. In addition, a large amount of purified VLPs of HEV-1, HEV-3 to 7 have been obtained.” (page 22, lines 439 to 443)
Overall, the manuscript is important for vaccine production progress and I would like to congratulate the authors for the good job.

Reviewer 3 Report
Comments to the author:
The manuscript by Bai et al. demonstrated a novel cell culture system for generating HEV virus-like particle and assessed the immunogenicity and antigenicity of the derived HEV VLPs. The study was interesting, but lack of novelty and there are several points need to be cited in this manuscript:
- As the authors mentioned, it is difficult to obtain large amount of viral antigen required for diagnosis and vaccine development, and the established system could generate 0.78 mg/107 cells (for RsVLPs) and 0.23 mg /107 cells (for RnVLPs) VLPs. I wonder it could be more convincing to compare the efficiency of this system to other established systems.
- The discussion part remains descriptive and lack of perceptiveness. The application, significance and advantages of this system should be more discussed.
- It is important to claim that the rabbits used were negative for anti-HEV IgG before immunized with RnVLPs or RsVLPs. Also, it could be interesting to see whether the immunogenicity and antigenicity of VLPs remains stable in larger number of animals. Minor points:
- The authors used HEV-1, HEV-3, HEV-4 and rat HEV, but no GenBank accession number was cited. It should also be cited that rabbits can be infected with different genotypes of HEV (Li S et al., Infectivity and pathogenicity of different hepatitis E virus genotypes/subtypes in rabbit model, Emerg Microbes Infect. 2020).
- ‘RsVLPs’ and ‘RnVLPs’ should be described before using abbreviation.
- Figure 5e: ‘Anti-rabbit’ HEV IgG.

Author Response
Responses to Reviewer 3:
Reviewers' comments:
The manuscript by Bai et al. demonstrated a novel cell culture system for generating HEV virus-like particle and assessed the immunogenicity and antigenicity of the derived HEV VLPs. The study was interesting, but lack of novelty and there are several points need to be cited in this manuscript:
- As the authors mentioned, it is difficult to obtain large amount of viral antigen required for diagnosis and vaccine development, and the established system could generate 0.78 mg/107cells (for RsVLPs) and 0.23 mg /107cells (for RnVLPs) VLPs. I wonder it could be more convincing to compare the efficiency of this system to other established systems.
Response:
We added the following sentences to the Discussion section: “Expression of the structural proteins by recombinant baculoviruses has long been used to generate VLPs of various viruses, and VLPs produced by this system usually retain the immunogenic and physicochemical properties of the native virions. In addition, a large amount of purified VLPs of HEV-1, HEV-3 to 7 have been obtained.” (page 22, lines 439 to 443)
- The discussion part remains descriptive and lack of perceptiveness. The application, significance and advantages of this system should be more discussed.
Response:
The significance and advantages of this system was added (page 22, lines 439 to 443).
- It is important to claim that the rabbits used were negative for anti-HEV IgG before immunized with RnVLPs or RsVLPs. Also, it could be interesting to see whether the immunogenicity and antigenicity of VLPs remains stable in larger number of animals.
Response:
The rabbits used for immunization were negative for anti-HEV IgG and HEV RNA. This information was added on page 7, line 155 in the revised manuscript.
Thank you for the suggestion regarding the stability of VLP immunogenicity/antigenicity. We would like to perform an experiment using a larger number of rabbits in the future.
Minor points:
4.The authors used HEV-1, HEV-3, HEV-4 and rat HEV, but no GenBank accession number was cited. It should also be cited that rabbits can be infected with different genotypes of HEV (Li S et al., Infectivity and pathogenicity of different hepatitis E virus genotypes/subtypes in rabbit model, Emerg Microbes Infect. 2020).
Response:
The GenBank accession numbers of HEV-1 (LC061267), HEV-3 (AB740232), HEV-4 (DQ079628), rabbit HEV (LC484431) and rat HEV (JX120573) used for the neutralizing assay were added at pages 9 to 10, lines 220 to 225.
Since this manuscript was not related to the infectivity of other hepatitis E virus genotypes in rabbits, we did not cite this reference.
- ‘RsVLPs’ and ‘RnVLPs’ should be described before using abbreviation.
Response:
Actually, ‘RsVLPs’ and ‘RnVLPs’ were not acronyms. However, we provided a description for VLPs upon first use.
- Figure 5e: ‘Anti-rabbit’ HEV IgG.
Response:
We corrected this. Thank you for catching it.

Round 2
Reviewer 1 Report
Although the authors did not provide convincing replies to some of the questions, I would accept the paper for publication.
Congratulations